# Assessing the Impact of Quarrying as an Environmental Ethic Crisis: A Case Study of Limestone Mining in a Rural Community

**DOI:** 10.3390/ijerph21040458

**Published:** 2024-04-09

**Authors:** Babalwa Kafu-Quvane, Sanelisiwe Mlaba

**Affiliations:** 1Faculty of Education, University of Fort Hare, Alice 5700, South Africa; 2Faculty of Applied Sciences, Cape Peninsula University of Technology, Cape Town 8000, South Africa; mlabas@cput.ac.za

**Keywords:** environmental ethics, crisis, quarrying

## Abstract

In this study, we investigated the impact of quarrying as an environmental ethical crisis. The need for the study arose when we realised the deteriorating effect of the quality of life in our community, which is located next to a limestone quarry. To obtain a deeper understanding of the adverse impact on the environment and the quality of life of the people living in the community around the quarry, we explored the workers from the community, and the members of the community-based organisation’s (CBO) experiences. We employed a qualitative method research approach, using a single case study design. We adopted a utilitarian perspective and Pinchot’s conservation as ethical systems that determine morality based on the greatest good for the greatest number. Both provide a framework for analysing environmental problems and ethical crises associated with limestone quarrying. We generated data using face-to-face interviews and focus group discussions. We present and discuss data through the following themes: analysis of the social and cultural impacts on local communities and indigenous people, assessment of the ecological consequences on biodiversity and habitat destruction, and examination of the effects on water resources, air quality, and soil erosion. The results show that the negative effects of the quarry on the environment have always worried the local people. The company’s disrespect for the community and ignorance of the laws governing quarry activities is the root of the ethical dilemma. The detrimental effects that the operations have on human health and safety as well as the environment is the other ethical dilemma, which includes, land degradation, vibrations, air, and water pollution.

## 1. Introduction

Limestone quarries provide various gains, such as societal, environmental, and economic. In addition to creating jobs and boosting the gross domestic product (GDP) of the nation, quarrying is essential to South Africa’s growth and development. It extracts a wide range of useful materials from the ground, such as coal, metals, stone, etc. Quarrying involves investigating potential sites of extraction, then getting the required material out of the ground, and finally blasting to get out the substance of interest. According to [1], limestone quarries provide raw materials which are used in the production of asphalt, concrete and other construction materials, which are used in the construction industry. Limestone is used in agriculture to adjust the pH level of the soil and to improve the soil quality. Furthermore, post-mining land in properly managed quarries is restored and rehabilitated for activities that include landscaping, reforestation, and local biodiversity preservation (conserving habitats and protecting endangered species. Quarries provide materials for construction projects, for roads and bridges, thus contributing to the local infrastructure, and thereby improving accessibility and connectivity in the surrounding area [2]. However, quarrying activities have a negative effect on the surrounding towns, the environment, and the health and safety of the workers and residents of the quarries such as landscape degradation, habitat destruction, air and water pollution, and noise pollution [3]. Despite the negative effects that quarrying has on the environment, human health, and safety, it is a welcome solution to difficulties for underprivileged people. There is a growing awareness among our societies and government officials that we run the risk of leaving future generations with a depleted environment if we carry on with our current level of environmental exploitation. As a result, our societies are creating community-based organisations (CBO) to support the government and business sectors in their environmental preservation efforts. Improving the standard of living for people who live in the areas surrounding quarries is the main goal of these CBOs. Our societies fully comprehend what [4] (p. 115) implies when he states that “the sustainability of the mining industry requires vigilance from everyone involved: governments, shareholders, communities, and industry”. In order to achieve a development that is socially just, environmentally sustainable, and economically efficient, mineral development must be completed in harmony with the environment, as recognised by the South African Constitution and its ensuing laws [5,6].

Our concern is motivated by the adverse environmental impacts and the effects that those impacts have had on the quality of life, of people, landscape restoration, and habitat reclamation.

## 2. Location and the Geology of the Area

Surrounding land use includes a natural ridge and valley to the south and west of the site, grazing and subsistence agriculture to the north, and the historical plant and stockpile area to the east (see Figure 1). The only major infrastructure in proximity to the site is the N2 highway which at its closest point is approximately 1.5 km to the east (see Figure 2). The closest receptors to the site are the school and some homesteads forming part of the Sierra Village located approximately 600 m north of the site. The other village is located approximately 1km over the ridge to the west (See Figure 2).

The closest major water courses are two rivers, one located 3 km to the east, and another river located 2 km to the west of the site. There is, however, a non-perennial drainage line in close proximity to the site (175 m to the north) into which any surface runoff from the site would flow. This drainage line then flows eastwards towards the Korana River (see Figure 1). The National Environmental Screening Tool of 2018, (Department of Forestry, Fisheries and the Environment, Republic of South Africa) identified very high sensitivities for terrestrial and aquatic biodiversity due to some of the sites falling within a critical biodiversity area according to the Eastern Cape Biodiversity Plan.

It is important to explore the mountains surrounding the area of study. The geological foundations of the Elundini Mountains, according to [7], in the Eastern Cape province of South Africa reflect a complex history of sedimentation, metamorphism, tectonic activity, and erosion, resulting in a diverse and visually stunning landscape (see Figure 3). The wider Drakensberg Range, of which these mountains are a part, is renowned for having a varied geological makeup. The oldest sedimentary and metamorphic rock formations make up the majority of them. Most of the underlying rocks are sedimentary in character, such as limestone, sandstone, and shale. These rocks evolved over millions of years because of sediment deposition in the former lake and marine environments. These sedimentary rocks underwent tremendous heat and pressure throughout time, which caused them to metamorphose into tougher, more resilient rock types such as quartzite and slate [8].

The area has seen tectonic activity, including folding and faulting, according to [8], which has led to the uplift and deformation of the terrain (see Figure 3). Numerous geological formations, including synclines and anticlines, are evidence of the region’s complicated tectonic past. In addition, the erosive powers of water over millions of years have moulded the Elundini Mountains’ untamed topography and deep river valleys. The landscape is still being shaped by this continuous erosion process, which also contributes to the distinctive geological features of the area by revealing various rock layers. This ongoing process of erosion continues to shape the landscape, exposing different rock layers and contributing to the unique geological features of the region.

## 3. Regulations Regarding Mining Activities

Regulations regarding mining activities vary greatly depending on the country. In South Africa, there is legislation governing mining activities to ensure the safety of workers, nearby communities, and the environment. To begin with, the constitution of the country stipulates that each individual has a right to an environment that is not harmful to their health or well-being, and to have the environment protected, for the benefit of present and future generations, through reasonable legislative and other measures that prevent pollution and ecological degradation; promote conservation; secure ecologically sustainable development and use of natural resources while promoting justifiable economic and social development [6].

When it comes to blasting methods in mining, safety zones are typically designated to prevent harm to personnel, equipment, and nearby structures. These safety zones are determined based on factors such as the type of explosives used, the size of the blast, the geology of the area, and local regulations [10]. Measures are implemented to prevent any pollution occurring during mining activities [11]. The waste generated as a result of the mining activities is disposed of appropriately [12]. The South African National Standards (SANS) are set for noise and dust generation during mining activities to ensure the health, safety, and sustainability of buildings [13].

## 4. Overview of the Use of Limestone in South Africa and Globally

Limestone is essential to the development of South Africa and the world economy, performing a variety of functions in a number of industries and promoting infrastructural development, economic expansion, and environmental sustainability [14,15]. Limestone deposits are abundant in South Africa, especially in the country’s northwest and northern parts. Because it provides raw materials for the production of cement, lime, and other industrial processes, limestone mining is important to the economy of the country. In South Africa, limestone is a key component in the creation of cement.

Major cement producers in the country rely on limestone quarries to supply the raw material for their manufacturing plants, which produce cement used in construction projects across the region. It helps enhance crop yields and supports sustainable agricultural practices. Limestone is utilised in various industrial applications in South Africa, including the manufacturing of steel, glass, chemicals, and paper products. It serves as a crucial raw material in these industries, contributing to their economic growth and development. Limestone is also valued for its aesthetic and historical significance in South Africa. It is used in the construction of heritage buildings, monuments, and landmarks, preserving the country’s architectural heritage and cultural identity [16].

According to research, limestone is a crucial component used worldwide in the manufacturing of cement and concrete. Building materials are made by crushing it and combining it with other materials. They are then utilised all over the world to construct roads, bridges, buildings, and other infrastructure projects. It is a source of calcium carbonate, which finds application in numerous industrial processes as a whitening agent, pigment, and filler. Limestone is used in environmental remediation projects, especially for wastewater treatment and acid mine drainage mitigation. Its alkaline qualities aid in the improvement of water quality in rivers, lakes, and industrial discharge sites by neutralising acidic pollutants. Because of its durability and aesthetic appeal, limestone is a popular material for countertops, floors, building facades, and decorative elements in architecture [15,17].

## 5. Background of Quarrying and Its Significance in the Mining Industry

Quarrying is a form of mining technology used to remove non-fuel and non-metal materials from rocks. The operation removes limestone, sand, and gravel from the geological bedrock. The minerals are extracted from the rock using a variety of procedures such as hard rock mining, which involves rock drilling, dynamite blasting, and other advanced ways [18]. Research shows that limestone is a calcareous rock that has at least 70% calcium carbonate content (CaCO_3_). Utilising limestone goes all the way back to prehistoric times when it was utilised by early humans to create weapons, construct homes, and construct furnaces. Taung in the Northern Cape is where South Africa’s limestone production began, deposits that were made up of a high-grade secondary limestone formation. An extensive geological search was launched in 1949 because it became evident that the rapidly expanding uranium industry would require large amounts of lime and the Taung reserves would not support such expansion [19,20]. A primary limestone close to Silver Stream was found as a result of the search. Throughout South Africa, the limestone industry has expanded ever since. Cement manufacturing, metallurgy (steel refining), agriculture (fertilisers, fungicides, livestock feed), aggregate, and lime processing are the five primary businesses in South Africa that employ limestone minerals. Cement manufacturing is the largest of these industries [20].

## 6. Explanation of Environmental Ethics and Their Relevance to Studying Quarrying Impacts

People hold different views about what is desirable or not, and their views not only form the basis of their attitudes towards the environment, but the ethical distinctions they make between what is morally wrong or right, good, or bad or what deserves respect or not. Acting morally is a key component of environmental ethics. Differing moral stances are what define arguments and disagreements over the state of the environment today. According to research, a lot of our attitudes toward nature have their roots in antiquated traditions that placed humans at the centre of the moral universe. According to [21,22], anthropocentrically oriented environmental ethics hold that the environment exists to give humans material satisfaction. Researchers use a ruthless developer as a metaphor to explain that humans maximally exploit nature in an unrestrained manner. For example, one of the impacts of quarrying is land degradation, whereby indigenous trees which are of economic and medicinal value are removed. This also disturbs biodiversity as certain habitats for some species are destroyed.

There are environmental ethicists with an ecological sensibility. They respect and treat the land accordingly as it is a community of life [22]. Biocentrism places the biological world at the centre of the planet and focuses on the intrinsic value of life. It does not include usefulness to human beings in its criteria. This value protects the land from the onslaught of mechanised man, hence [23] states that something is good when it preserves the integrity, stability, and beauty of a biotic community. It is wrong when it tends otherwise. Quarrying destroys the entire local ecosystem. The gains made from quarrying, like providing job opportunities and impacting the economy are all short-term. The preservation of vegetation, animals, and the overall ecosystem needs to be considered as well. The authors of [22] mention the intrinsic value of the beauty of surroundings, and nature itself. Approaches to morality can justify attitudes and principles that if widely adopted would lead to a less destructive and more respectful treatment of nature. Maintaining such an environment can be valuable within even a human-centred ethical framework since humans require an environment in which they can thrive [22]. Our world heritage, the wildness that we inherited from our ancestors, needs to be appreciated and protected in order for future generations to have it all. From a sustainability perspective, this does entail that any such rationale needs to properly include the worth of nature (natural beauty) for future generations. Natural environments are destroyed by quarrying, depriving future generations of their beauty. The economic growth brought about by quarrying comes at a great cost that will be borne by future generations.

## 7. Introduction of the Case Study: Limestone as a Prominent Quarrying Material

Sierra is a little settlement in the Eastern Cape province in South Africa (see Figure 1) that is part of the OR Tambo Municipality. There are just 150 people that live in this isolated community. The primary economic activities in Sierra for the longest time were animal husbandry and subsistence farming, which were made possible by the area’s micro forest and perennial river. A young man who was born and reared in Sierra came up with the plan to open a limestone quarry in the community to promote both social and economic advancement. He identified a gap in the local cement production industry, so he came up with the concept to blast and extract limestone so that people could:Purchase raw limestone locally.Make bricks for local consumption.Produce cement.

The young man explained his plan and assured them that if the quarry was a success, local people would be given priority when it came to jobs there. They would once again be given priority when they ascended to more senior roles. In addition to infrastructure development, the community was promised a skill transfer program to address the most apparent skill gaps in the area. The community members gave the young man permission to move forward with his planned development because they regarded it as a viable answer to one of their issues—the high rate of youth unemployment. The next concern is how to extract minerals to expand our economy without endangering the environment or endangering human health and safety. This study uses the conservationist ethic of Gifford Pinchot, sometimes known as Pinchot, to address this question.

## 8. Overview of the Existing Literature on Quarrying Impacts and Environmental Ethics

According to the South African Mineral Resources Government’s Annual Report, 2017, the infrastructure development connected to South Africa’s quarrying mines employed over 471,000 people overall in the country’s fourth quarter of 2015. This demonstrates the significance of the quarrying industry to the economy of the nation as well as to the welfare and means of subsistence of nearby communities [24,25]. But quarrying devastates the environment in ways like shifting the soil, making it infertile, changing the landscape, losing plants and vegetation, turning hillsides into unsightly scars, deteriorating the ecology and aesthetics, lowering the water table, changing, or eliminating natural drainage systems, and polluting the environment [26,27,28].

According to [24], these quarries’ dust has posed health and environmental risks. In order to identify the consequences of quarrying on the environment, a health risk assessment of the areas surrounding quarries in Southwest Nigeria was conducted. During blasting operations and product transportation, quarrying sites generate dust, which contributes to air pollution. The climate of the region, the amount of dust in the air, the kind of rocks being mined, and the chemical composition of the dust all affect the surrounding ecosystem [29,30]. Dust has a direct effect on animals and plants, resulting in lung problems in animals and poor growth in plants. However, noise pollution has been the most common type of pollution brought on by quarrying operations. Almost every aspect of the environment—landscape, water, soil, subsoil, flora, fauna, etc.—has been touched by gypsum extraction, mainly the karstic forms. Not only does it threaten the caverns and the natural resources they hold, but it also impacts the surface waters, significantly altering the hydrographic network. Additionally, groundwater, which is currently found at the bottom of a quarry, is also affected [31].

## 9. Exploration of Key Concepts: Sustainability, Biodiversity, and Land Degradation in Relation to Quarrying

### 9.1. Sustainability and Quarrying

Research shows that quarrying is faced with many challenges due to sustainability issues and environmental concerns. The quarrying mines’ poor management and operations have had a severe negative influence on the environment. [32,33,34]. Rock drilling, blast, and other sophisticated methods that are used to extract minerals from the rock may result in land use change, leading to excessive noise, deforestation, soil erosion, alteration of soil profiles, contamination of local streams and wetlands, and dust [34]. From a sustainability perspective, the value of natural beauty for future generations must be fully considered. Generations to come will no longer be able to enjoy the beauty of natural surroundings due to quarrying. Quarrying promotes economic expansion, but at a great cost that will be borne by future generations.

### 9.2. Biodiversity and Quarrying

Research shows that open mining has a devastating negative impact on the surrounding environment situated nearer the quarry. Activities that are performed result in ecological destruction, destroying natural habitats. Heavy metals are well-known industrial pollutants and have a toxic effect on animals and plants. They accumulate in the food chain and are deadly harmful to both animals and plants [3,35,36,37]. However, quarries, gravel pits, and other open-pit mines can be very beneficial for environmental conservation since they provide untouched habitats for endangered and protected species, according to a number of scientific studies [34,38,39]. Quarries and gravel pits provide a vital home for plants and animals that are being driven out of other areas by development.

### 9.3. Land Degradation in Relation to Quarrying

Environmental damage from quarrying activities includes soil displacement, landscape change, and soil infertility. This has been observed to have significant effects on the environment and agricultural industry [1]. A study by [33] found that some places may be subject to soil erosion caused by quarrying operations, which results in stormwater runoffs. It has been difficult for most African communities living near quarrying mines because water runoff floods farmlands, and in some cases, these mines pump water from open pits into the river or stream that the community depends on for daily needs and agricultural practices. The humus in the soil, which is necessary for plant growth, is destroyed when surface vegetation is cleared away for quarrying.

## 10. Discussion of Various Theoretical Frameworks Used in Assessing Environmental Ethics in Quarrying

There are different theories that stress the morality that acknowledges the importance of nature. According to the utilitarian perspective, a decision is morally correct if it maximises the good for the greatest number of individuals [40]. This perspective encourages maximising benefits for humans at the expense of the environment. The greatest good for the greatest number is its guiding principle. This theory’s contribution to the field of environmental ethics can be attributed to the fact that it acknowledges that some distant outcomes or anticipated repercussions do matter in moral thinking [41]. Secondly, the highest moral standard, according to [42] is to maximise human enjoyment while reducing suffering. The environment is essential in the context of human happiness. The preservation of an environmental life support system is necessary for human satisfaction. Beyond the bare necessities of life, medicine, economic gains, artistic pleasures, and other natural resources with useful qualities are what make people happy. An extended anthropocentric theory claims that humans have a responsibility to protect the environment, protecting the environment is necessary for the welfare of coming generations. Concern for the environment is ultimately for the benefit of the next generation of humans.

The utilitarian theory is in favour of the intergenerational equity principle. There is a concern for future generations, which is founded on the intergenerational equity principle, according to [40]. The benefits of nature must be enjoyed by future generations in the same ways that we as the current generation do. This implies that there needs to be a balance between our personal interests and the interests of our descendants. Serving nature is helping the next generation of humans. The long-term consequences of environmental exploitation, including quarrying, marine pollution, etc., will be felt most keenly by humans for many generations in the future. Quarrying is important in South Africa since it helps the country’s local economic development. In addition, using the materials that are extracted improves trade and opens job opportunities. According to the Department of Mineral Resources 2017 report, the market study of the construction sector revealed an exponential rise in the actual value added to the GDP during the previous ten years. Human-centred ethics that are solely focused on the advantages of the current generation, such as quarrying, which gives greater emphasis to the country’s development and more economic growth, will result in unequal development. This results in growth at the expense of environmental destruction. The impact of it though may not be felt by the present generation its evil effects will be most acutely observed in the lives of future generations.

Detailing the utilitarian ethical framework, ages ago, Bentham Jeremy, declared that the utilitarian ethical framework establishes morality based on the greatest good for the greatest number [43]. We can describe Pinchot’s conservation in terms of the utilitarian system based on this description. Conservation, in my opinion, might help solve the environmental issues brought on by limestone quarrying. Pinchot lays a solid basis for his conservation ideas, which opened the door for other laws aimed at preserving the environment. Encouraging a better planet for ourselves and our future generations to live in is the main goal of the conservation movement. Pinchot thought it was crucial to resolve conflicts of interest by making decisions based on what would be best for the greatest number of people over the long term. It opposes the wasting of natural resources and, more significantly, it advocates for granting everyone the same access to resources and the advantages they provide. Pinchot defined conservation as “applying common sense to common problems for the common good” [44] (p. 72).

## 11. Research Methodology

An investigation was conducted on a case study of a limestone quarry in the Eastern Cape, located in the small community of Sierra. Purposive sampling was used in this investigation. Purposive sampling involves the researcher intentionally choosing research participants based on how well they will fit the study’s goals. Research shows that purposive sampling is carried out with a specific goal and based on the researcher’s judgment, either on the ownership of the characteristics sought for an in-depth study or on the specific selection or handpicking of information-rich cases [45,46,47,48]. Purposive sampling is generally utilised to choose appropriate examples and skilled individuals who possess an in-depth understanding of the problem at hand.

We handpicked the quarry based on its relevance to the problem we were exploring. Through purposive sampling, we selected eight workers who were from the surrounding communities and five members of the community-based organisation (CBO). The participants were a mixture of both genders, and their ages were varied. They were chosen because they were a good source of information for the study.

Focus groups, also known as discussion groups or group interviews, are popular in social research for using group interview settings for data generation. They are open discussions between the researcher and the research participants, exposing the researcher to different perceptions that the participants hold about a particular topic [46]. Focus group interviews with the workers and semi-structured interviews with the members of the community-based organisation were major tools to generate data in this study. Case studies benefit from having multiple sources of evidence [49]. We used semi-structured interviews with the members of the community-based organisation because they allowed us to ask follow-up questions for clarity. By doing so, the trustworthiness of what we were looking for was achieved.

To create intelligible accounts of data we used a process of data analysis, which involves categorising and re-combining evidence to produce empirically based findings [37,49]. Data analysis, according to [49], is the process of tabulating, analysing, classifying, testing, and recombining evidence to generate conclusions with an empirical foundation. It is possible to create understandable accounts of data through the process of data analysis. According to [46], qualitative researchers use data analysis to interpret their findings by looking for patterns, themes, categories, and regularities.

## 12. Quarrying Impact on the Environment: Presentation of Empirical Evidence on the Environmental Impacts of Quarrying Activities

In exploring the impact of limestone quarrying on the environment, the following questions were discussed:What sociocultural impacts did the quarry have on the people of surrounding communities?What environmental impacts did the quarry have on the communities around it?

The following subthemes emerged from the data: analysis of social and cultural impacts of limestone quarries on communities and their people, assessment of the ecological consequences on biodiversity and habitat destruction, and examination of the effects on water resources, air quality, and soil erosion. The workers were interviewed separately from the members of community-based organisations. We wanted their views as they were directly involved in the quarry operations.

### 12.1. Analysis of the Social and Cultural Impacts on Local Communities and Indigenous People

Data shows that people living close to the quarry are more vulnerable to the devastation produced by quarrying operations. Despite the beneficial effects on the local and national economies, the local populations nonetheless confront a variety of significant issues as a result of the quarrying mines [50]. Research illustrates how mining activities impact the well-being of local communities and the environment. One such scenario involves the interaction between mining enterprises and indigenous populations of the Arctic region, particularly in the extraction of placer diamonds. Ethnological expertise informs these projects, emphasising considerations such as the indigenous community’s interests, compensation for damages incurred, support for traditional crafts, and employment opportunities. In another study, gender dynamics within indigenous communities in the Arctic are explored concerning their involvement in mineral extraction projects. This research reveals that indigenous women prioritise environmental preservation and safeguarding their living conditions, while men are more focused on economic aspects such as income and employment opportunities [51,52]. Beneficial effects of the quarry include giving young people work opportunities and fostering local economic growth. Community members, however, are extremely unhappy with the negative effects and believe the project’s promise of a better life was a scam. When mining firms disregard the community or fail to include them in development initiatives aimed at raising living conditions in the areas where they operate, the surrounding communities typically experience actual problems of the mines rather than just imagined ones.

Data reveal that people in Sierra, especially those who lived close to the quarry, started to notice that their homes were becoming cracked over time. For individuals who remained to farm for subsistence, there was a decrease in annual harvest and a reduction in plant growth, along with a rise in respiratory ailment incidence. Along with the high prevalence of waterborne illnesses among the locals, they also observed that the majority of the animals in the village that drank from the perennial river were certain to become ill. Therefore, the list of detrimental consequences on the community’s environment that have resulted from the quarry’s operation includes degradation of the biota and habitat, influence on the quality of the water, and contamination of the air and land.

### 12.2. Assessment of the Ecological Consequences on Biodiversity and Habitat Destruction

According to the members of the community-based organisation, a portion of the tiny forest was destroyed early in the project to create space for the building of shops, restrooms, offices, and workshops. Because native trees had both commercial and therapeutic uses, their removal led to land deterioration. The loss of trees also disrupted biodiversity since it eliminated some species’ habitats. The engineering processes involved in blasting and extraction are directly responsible for the environmental disruption produced by quarrying. The participants identified several clear effects of quarrying, including habitat loss, noise, dust, vibrations, and erosion that alter the geomorphology of the area and modify its use, changing the surrounding visual environment. Although there are other types of machinery used in quarries that cause vibrations, blasting is thought to be a primary contributor [34]. Data reveals that at Sierra, blasting and drilling are used to extract minerals. The people who live near the blasting zone connect the blasting to the cracks in their homes. They added that a few stone-throwing events have also occurred during the blasting, which has led to a few mishaps like glass shattering in some homes. Karst topography has considerable scenic value, as noted by [53] a century ago, which amplifies the visual impact of quarrying. They argue that the removal of stones, which is the principal geomorphic impact of quarrying destroys habitats including relics and active caves.

According to [54], caves develop one of the most peculiar terrestrial ecosystems. Such communities are determined for life by the lack of light, and based on the degree of darkness, such environments are divided into the following zones; twilight zone, where various fauna are found; the transition zone of complete darkness where common species which make sorties to the outside world; a zone of deep darkness with full cave-adapted species, and the stagnant zone of darkness, with little air exchange and high concentrations of carbon dioxide. Numerous bat species, such as those that feed on nectar and others that eat insects, spend their nights in the twilight or transition zones of caves. The world’s largest known animal colonies are made up of insectivorous bats [55]. These zones are also home to plants, animals, and birds. The habitat that caves provide disappears as rocks are removed by quarrying, killing off species that have adapted to the deep, stagnant zones mentioned above.

### 12.3. Examination of the Effects on Water Resources, Air Quality, and Soil Erosion

The participants’ responses show that dust polluted the air, and this has been the situation in Sierra Village since the project’s inception and the inhabitants have several incidences of respiratory illnesses. They also reported that blasting activities disrupt the groundwater and surface water flow, which lowers the amount and quality of drinking water available to them and wildlife downstream from the quarry site. Water pollution can also be caused by improper waste management. The participants also reported that prior to the development and operation of the quarry, the Sierra community recognised that there were issues with the amount and quality of water. However, the quarry has made things worse because animals are now dying, and the community members must boil and purify water before drinking it to prevent waterborne illnesses.

Ref. [56] is of the view that dust is a major cause of air pollution, the intensity of which varies depending on the local climate, the amount of dust particles in the surrounding air, the size of the dust particles, and their chemical makeup. For instance, dust from limestone quarries is primarily alkaline. In addition to its propensity to cause respiratory problems and surface deposition, air pollution can also physically harm plants by abrading their leaves and obstructing or destroying their interior structures [56]. Surface water flow is altered by quarrying-related engineering activity [35]. Blasting alters the flow of groundwater, which in turn alters the flow of surface water. Therefore, releasing quarry water into neighbouring streams increases the time between flood occurrences. The dissolution and settling of dust in water bodies is the source of water pollution in the vicinity of limestone quarries.

## 13. Results

### The Environmental Impact of Quarrying as an Ethical Crisis

Residents in the community have been continuously concerned about all the negative environmental effects of the quarry. The company ignored the neighbourhood and did not know about the laws controlling quarry operations, which led to an ethical dilemma. The harm that the operations cause to people and the environment is the other ethical dilemma. Being a utilitarian who was greatly impacted by Pinchot’s conservationism, I firmly believe that the project’s execution was immoral since it was unable to provide “the greatest good for the greatest number of people.” The project did not improve the community members’ quality of life; rather, a small group of wealthy people profited from it constantly. The project also resulted in abandoned quarries with disturbed vegetation and a loss of biodiversity, which is against all conservationist principles. A group of community members came together to form a community-based organisation called “Ikamva-lethu,” which means “our future,” since they firmly believed that the project needed to be more inclusive and supportive of the community. Ikamva-lethu sought to discuss the best ways to address the community’s problems with quarry management. The quarry CEO revealed fresh information a few weeks prior to the start of the engagements. The quarry was being transferred to GT as part of a corporate rescue plan because the quarry company had already racked up ZAR 200 million in debt as a result of losses and lawsuits. Nevertheless, there was a bright spot to those depressing reports: GT management was eager to collaborate with CBOs to improve environmental preservation. After that, Ikamva-lethu was given the assignment to draft suggestions, which they were to provide to the incoming management after the engagements had started. One of the key principles of Ikamva-lethu is that the development must involve the community and the quarry must function and grow in harmony with the surrounding ecosystem.

## 14. Recommendations Using Principles of Conservationism

Ikamva-lethu’s recommendations to the new management are all well-guided by the fundamentals of conservationism. The authors of [57] distilled the concepts of conservationism into three main points: distribution of resources to all people, prevention of waste, and development. Pinchot was adamant about growth because he thought that conservation necessitated both forward-thinking planning and prioritising the rights and needs of the current generation. This idea, in my opinion, is right in line with what Ikamva-lethu stands for. They are not opposed to mineral extraction, but they do require that quarrying be conducted in a way that is environmentally friendly and takes into account the health and safety of the public as well as the natural environment. Planning and executing correctly will be the most effective way to do this. The authors of [57] recognised the finite nature of resources and the need to exercise caution to avoid squandering them. To ensure that operating procedures are planned sustainably and that less waste is generated, Ikamva-lethu needs to bring this fact to the attention of the new management.

Pinchot was a fervent supporter of equitable resource distribution and did not only believe in development; rather, he believed in development that benefited the greatest number of people. In my opinion, the quarry fell short in this regard; years after the project started, not a single community member owned a part of the quarry. Ikamva-lethu and GT management must collaborate closely to develop plans for enabling the fair distribution of minerals in the quarry, whether in the form of shares or royalties. Pinchot thought it was crucial to resolve conflicts between interests by making decisions based on “the greatest good for the greatest number in the long run” [44].

Adopting conservationism, in my opinion, would help to guarantee that community members’ grievances are addressed and that more people will ultimately profit from the initiative. Most criticisms levelled at conservationism claim that it is overly anthropocentric. While I agree that conservationism frequently has a greater concern for human welfare, it also acknowledges that society and nature are interdependent, that human welfare depends on the welfare of nature, and that nature occasionally needs human interaction to maintain the integrity of its evolutionary cycle.

## 15. Incorporating Conservationism to Mitigate the Negative Environmental Impact

The quarry project prioritised the extraction and exploitation of minerals, with little regard for the project’s effects on human health and safety or the environment. Since the project’s inception, the environmental authorisation requirements have not been strictly followed. However, the Sierra community is aware of what [4] (p. 11) means when he states that “vigilance from everyone involved is required for a project to succeed.” They are prepared to collaborate with the developers to make the quarry project a success and to guarantee that the development will not negatively impact their environment, public health, or safety. They understand that environmental conservation is the responsibility of all parties. According to the chairperson of Ikamva-lethu, the developers’ lack of environmental planning and consideration during the project’s early phases is to blame for several issues related to the quarry. He went on to say that the community wanted GT to take responsibility for all the social evils the quarry firm had created over the years by meticulously planning all of its operations and putting in place stringent mitigation strategies and appropriate monitoring equipment. They anticipate that the new management will address the problem of skills transfer and that GT will implement a socioeconomic program aimed at empowering the community and giving them a stake in the quarry.

Following the guidelines of conservationism, Ikamva-lethu needs to properly inform the GT management of the following corrective measures:Correct planning of the corrective measures is necessary. The corrective measures should take precedence over sustainable development and concentrate more on methods to stop the degradation of landforms, habitats, garbage, etc.To guarantee that the project will please developers and community members and achieve “the greatest good for the greatest number of people in the long run,” common grounds for decision-making for each challenge must be developed.To ensure that the interests of the community and developers are aligned over the long term, platforms must be established to maximise community involvement.

Taking into account Sierra’s current situation, the following suggestions and mitigating measures are offered; it is advised that the new administration incorporate them into their EMP.

Research should be conducted to find out what can be completed to restore the environment, such as revegetating old quarries and using restoration blasting—a carefully planned method of blasting carbonate rock quarries that, according to [58], can create buttresses, talus slopes, and headwalls that can be revegetated to produce plant assemblages and landforms that resemble those that naturally occur on valley sides. Examining the effects of gold mining on both the economy and the environment, a case study in Ecuador illustrates efforts to mitigate past environmental damage. Notably, former mine sites are evaluated for their potential in tourism development, aiming to generate employment opportunities and income for local residents, while also addressing existing environmental and social challenges [59].Suggest moving those who live near the blast zone; doing so will protect their health and safety by removing the possibility of homes falling and hurting families. The Constitution of South Africa Act, and the Interim Protection of Informal Land Rights Act, must be followed if residents living near the explosion zone consent to the relocation. These legislative clauses guarantee that the affected groups receive just and equal treatment and that no one will be in a worse situation than they are in at the moment.In order to remove vibrations, it is necessary to monitor the vibrations and notify the community members when blasting is scheduled.Installation of dust control devices, such as water sprinklers, at the unit level.Use geographic information systems (GIS) to create a framework for assessing water quality and a standard syntax for it. To give decision-makers using water resources a solid foundation, these systems need to facilitate forecasting of water quality.Investigate waste prevention techniques to ensure that generated by-products do not wind up in water sources; this will also help to preserve scarce resources.Increase community involvement, especially when it comes to environmental conservation; to do this, community trust needs to be built, and measures need to be taken to guarantee that a share of the quarry’s ownership is given to locals.

## 16. Conclusions

Measures for the extraction of minerals should be approached comprehensively, harmonise the interests of all the people involved, from the standpoint of environmental ethics, taking into account the social and moral aspects of people’s lives, their culture, ethnicity, traditions, etc. In this case, moral behavior is a key component of environmental ethics.

The negative impacts of limestone quarries need to be carefully managed through proper planning, regulation, and implementation of best practices to maximise the gains while minimising the adverse effects on the environment and local communities. Planning with the future in mind can help minimise a quarry’s environmental impact during its operational phase, according to [60]. 

## Figures and Tables

**Figure 1 ijerph-21-00458-f001:**
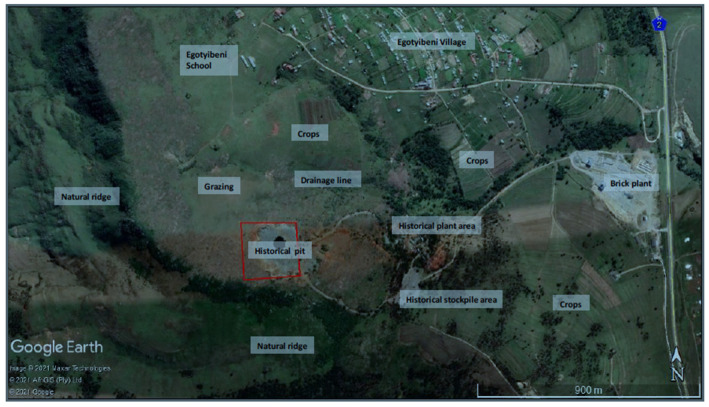
Land use and environmental sensitivity.

**Figure 2 ijerph-21-00458-f002:**
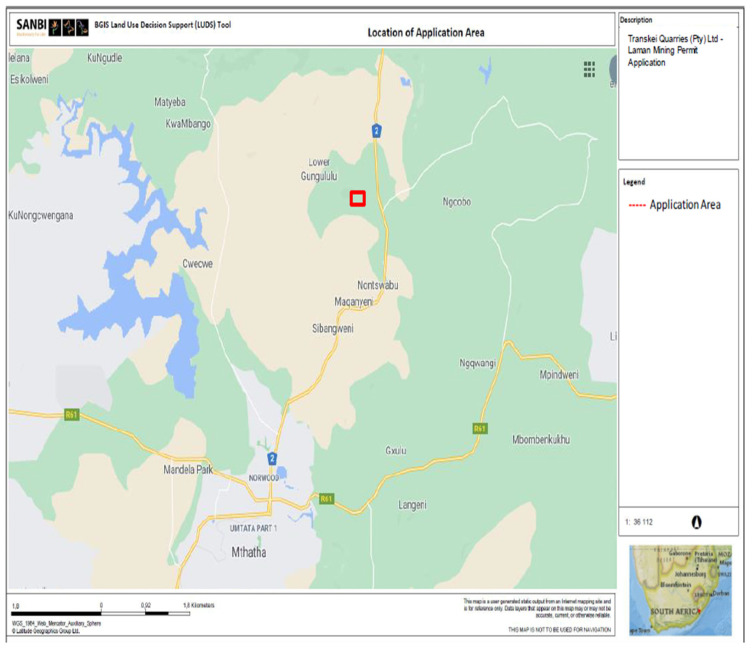
Locality map.

**Figure 3 ijerph-21-00458-f003:**
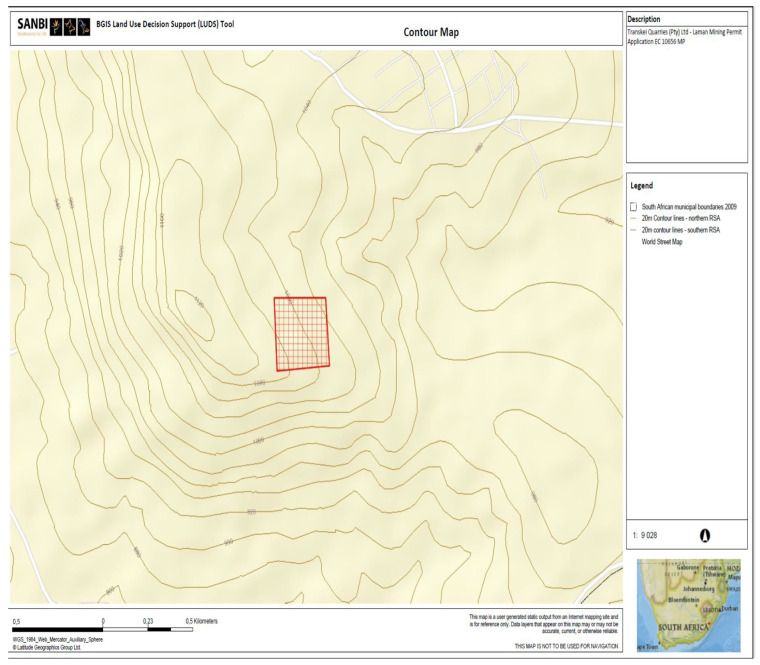
Map showing contour lines indicating the topography of the site and surrounding areas [9].

## Data Availability

The data are unavailable due to privacy or ethical restrictions.

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
