# Peer review of "Assessing the Impact of Quarrying as an Environmental Ethic Crisis: A Case Study of Limestone Mining in a Rural Community"

_ijerph, 2024, doi:10.3390/ijerph21040458_

Round 1
Reviewer 1 Report (Previous Reviewer 1)
Comments and Suggestions for Authors
Good revision
Author Response
Thank you.
Reviewer 2 Report (Previous Reviewer 2)
Comments and Suggestions for Authors
The text has been well improved and enriched with some new information.
Author Response
Thank you. I have worked on the document again, fixed little errors, and worked on the reference list.
Reviewer 3 Report (New Reviewer)
Comments and Suggestions for Authors
The topic of the article is relevant. The paper examines the issues of assessing the impact of career work from the standpoint of environmental ethics and the impact on the environment and living conditions of the population in rural areas. This approach has important scientific and practical significance. Using a simple and understandable example of limestone mining, the possible consequences of such activities for the economy (cement production, house construction, etc.), as well as for the environment and environmental living conditions of the population (transformation of the landscape, reduction of biodiversity, deterioration of aesthetic properties of the territory, etc.) are analyzed. The article uses modern research methods (qualitative method, ecological and economic analysis, conducting interviews and discussion in focus groups. Particularly important is such an aspect of the article as the ecological, economic and social assessment of the considered activities on the indigenous population from the standpoint of the impact on the health of the local population, land degradation, air and water pollution.
The positive aspects of this article include the fact that the presented material is considered broken down into small sections, characterizing, for example, the impact of a limestone quarry on obtaining various benefits (social, environmental and economic), including job creation, the opportunity to generate income for the local population. From this perspective, as the authors rightly point out, the use of natural resources is essential for the growth and development of South Africa. On the other hand, it has been shown that limestone mining has a negative impact on land resources, pastures and subsistence agriculture of the local population, and can also lead to environmental changes in the area under consideration as a whole. It should be noted that some scientific statements and research results are well illustrated with examples, drawings, and maps.
The article analyzes legal and other regulatory measures for the extraction of minerals in order to develop mechanisms for harmonizing the interests of cement producers, the local population and to prevent environmental pollution. An important conclusion of the article is that the development of such maps should be considered in a complex from the standpoint of environmental ethics, i.e. not only from the utilitarian standpoint of economics or only from the standpoint of environmental conservation, but also taking into account the social and moral aspects of people's lives, their culture, ethnicity, traditions, etc. In this case, moral behavior is a key component of environmental ethics. The authors critically analyze in detail the available approaches in this area. It should be noted that during the preparation of the article, the authors worked through 51 sources of literature.
Based on the analysis of the social and cultural impact on local communities and indigenous populations, environmental impact assessments, and biodiversity, the authors formulate specific recommendations and corrective measures that take into account the interests and needs of all stakeholders. Thus, the authors consider the impact of mining on the environment as an ethical problem, as well as a problem of ensuring environmental and social conditions for the quality of life of the local population. This implies the inclusion of these issues in the project analysis system, i.e. for the success of the project, it is necessary to take into account all aspects of the development of this territory, including planning, development and implementation of corrective measures, informing the population, creating a platform for the involvement of the local community, etc.
As for the comments and suggestions to the authors, we can suggest highlighting other similar examples in other regions (or at least making references to literature) when mining affects the interests of the local population and the environment. For example, at work Novoselov A., Potravny I., Novoselova I.Yu., Gassy V.V., Sharkova A.V. Harmonization of interests during arctic industrial development: the case of mining corporation and indigenous peoples in Russia. Polar Science. 2023. Vol. 35. P. 100915. The issues of interaction between mining companies and indigenous peoples of the North in the extraction of minerals (placer diamonds) in the Arctic are considered on the basis of ethnological expertise of projects, which involves taking into account the interests and needs of the local population, compensation for damage caused, support for traditional crafts of indigenous peoples, creation of jobs for the population, etc.
In Potravnaya, E., Hye-Jin, Kim (2020) Economic Behavior of the Indigenous Peoples in the Context of the Industrial Development of the Russian Arctic: A Gender-Sensitive Approach. REGION: Regional Studies of Russia, Eastern Europe, and Central Asia, 9(2), 101-126. DOI: 10.17516/1997-1370-0780 EDN: HQSYQK analyzes the gender aspects of indigenous peoples' behavior in the justification and implementation of mineral development projects in the Arctic. This study shows that indigenous women are more concerned about environmental conservation and the preservation of their environmental living conditions, while men are more concerned about income, employment, etc.
At work Apolo Herrera A. E., Chavez Ferreyra K.Y, Potravny I.M. Gold mining impact assessment on the economy and the environment in Ecuador // "GORNYI ZHURNAL", 2020, No. 2, pp. 62-65. DOI: 10.17580/gzh.2020.02.08) (in Russ.) The impact of gold mining on the economy and the environment is analyzed using the example of Ecuador in the context of eliminating past (accumulated) environmental damage. It is noteworthy that the territories of former mines and gold mines are considered from the point of view of tourism development, which creates employment and income conditions for the local population, allows creating conditions for solving existing environmental and social problems. In general, the reviewed article meets the requirements, the results of the study indicate an increase in new scientific knowledge.
In view of the above, the article is recommended for publication.
Author Response
I highlighted instances (from three countries) illustrating how mining activities impact the well being of local communities and the environment.
See pages 11, 12 & 16) They are highlighted in yellow. Literature is referenced and highlighted in yellow.
This manuscript is a resubmission of an earlier submission. The following is a list of the peer review reports and author responses from that submission.
Round 1
Reviewer 1 Report
Comments and Suggestions for Authors
In this study, the impact of quarrying as an environmental ethical crisis was investigated. This study is in line with the scope of the journal. I think it is an interesting work. However, the research methods in the paper are too qualitative. Not even a single image or table. I suggest the author to conduct some quantitative research and add some images or tables to make the article more academic.
Comments on the Quality of English LanguageMinor editing of English language required
Author Response
.

Reviewer 2 Report
Comments and Suggestions for Authors
The text is very general and the list of publications -incomplete, especially with the regard to the assessment of the environmental impact of exploitation and remedial measures. Conclusions and suggested actions are obvious. It is necessary deep revise of the text and complete it in the following areas:
1. description the location of the quarry and its infrastructure in relation to the residential buildings, roads, and other elements of land cover (areble fields).
2. description of the applicable regulations regarding mining activities, in particular designation of safety zones when using blasting methods. Whether such zones are required and where they destignated in the presented case. Are there houses within them?
Such requirements are common in most countries and regulated in frame of the mining licence. In case of violation, the relevant mining supervision bodies break the operations of the mine. Similar situation is in the case of violation of other formal and environmental requirements. Such reference is missing in the text.
3. Enlargement the methodology description - especially presenting the set of questions asked to the respondents or the scope of survey, as well as the size of the respondents group. The questions should be the same for all and allow for clear conclusions. The group of intervieved people should be representative. Intervievs with the local people should not only consist in listening to their complaints. The research despite its qualitative nature must be comparable and measurable.
It also seems that the quality of the English language is not sufficient. Moerover it is suggested in the text, that the research was carried out by one person, while there are two authors of presented issue.
Author Response
.

Reviewer 3 Report
Comments and Suggestions for Authors
Please check the attached file.

Author Response
.
